# Natriuretic Peptides in Heart Failure with Preserved Left Ventricular Ejection Fraction: From Molecular Evidences to Clinical Implications

**DOI:** 10.3390/ijms20112629

**Published:** 2019-05-28

**Authors:** Daniela Maria Tanase, Smaranda Radu, Sinziana Al Shurbaji, Genoveva Livia Baroi, Claudia Florida Costea, Mihaela Dana Turliuc, Anca Ouatu, Mariana Floria

**Affiliations:** 1Department of Internal Medicine, “Grigore T. Popa” University of Medicine and Pharmacy, 700111 Iasi, Romania; tanasedm@gmail.com (D.M.T.); sanziana.alshurbaji@yahoo.com (S.A.S.); ank_mihailescu@yahoo.com (A.O.); floria_mariana@yahoo.com (M.F.); 2Internal Medicine Clinic, “Sf. Spiridon” County Clinical Emergency Hospital Iasi, 700115 Iasi, Romania; 3Cardiology Clinic, “Prof. Dr. George I.M. Georgescu” Institute of Cardiovascular Diseases, 700503 Iasi, Romania; 4Institute of Gastroenterology and Hepatology, 700115 Iasi, Romania; 5Department of Surgery, “Grigore T. Popa” University of Medicine and Pharmacy, 700111 Iasi, Romania; hauliviagenoveva@hotmail.com; 6Vascular Surgery Clinic, “Sf. Spiridon” County Clinical Emergency Hospital Iasi, 700115 Iasi, Romania; 7Department of Ophthalmology, “Grigore T. Popa” University of Medicine and Pharmacy, 700115 Iasi, Romania; costea10@yahoo.com; 82nd Ophthalmology Clinic, “Prof. Dr. Nicolae Oblu” Emergency Clinical Hospital, 700115 Iași, Romania; 9Department of Neurosurgery, “Grigore T. Popa” University of Medicine and Pharmacy, 700115 Iași, Romania; turliuc_dana@yahoo.com; 102nd Neurosurgery Clinic, “Prof. Dr. Nicolae Oblu” Emergency Clinical Hospital, 700115 Iași, Romania

**Keywords:** heart failure, natriuretic peptides, preserved ejection fraction

## Abstract

The incidence of heart failure with preserved ejection fraction (HFpEF) is increasing and its challenging diagnosis and management combines clinical, imagistic and biological data. Natriuretic peptides (NPs) are hormones secreted in response to myocardial stretch that, by increasing cyclic guanosine monophosphate (cGMP), counteract myocardial fibrosis and hypertrophy, increase natriuresis and determine vasodilatation. While their role in HFpEF is controversial, most authors focused on b-type natriuretic peptides (BNPs) and agreed that patients may show lower levels. In this setting, newer molecules with an increased specificity, such as middle-region pro-atrial natriuretic peptide (MR-proANP), emerged as promising markers. Augmenting NP levels, either by NP analogs or breakdown inhibition, could offer a new therapeutic target in HFpEF (already approved in their reduced EF counterparts) by increasing the deficient cGMP levels found in patients. Importantly, these peptides also retain their prognostic value. This narrative review focuses on NPs’ physiology, diagnosis, therapeutic and prognostic implication in HFpEF.

## 1. Introduction

The incidence of heart failure (HF) is increasing. If for HF with reduced ejection fraction (HFrEF) there are well-established methods of diagnosis and treatment, this is far from true in HF with preserved ejection fraction (HFpEF) patients. This increasing incidence justifies the need for proper diagnostic, therapeutic and prognostic tools. In the era of cardiac biomarkers, natriuretic peptides (NPs) have a well-established role in HF pathophysiology and patient management. However, both the lack of consensus regarding NPs in HFpEF and the heterogeneous population makes diagnosis and management difficult and unstandardized, leading to a decrease in quality of life and increase in mortality and hospitalization.

## 2. Heart Failure with Preserved Left Ventricular Ejection Fraction

Heart failure represents a clinical syndrome affecting 2%–3% of the general population [1]. Its prevalence is increasing and it leads to an increased risk of death, increased hospitalization rates, a decrease in quality of life and higher costs through complex therapeutic strategies [2,3].

One of the classifications of HF is made by determining the left ventricular ejection fraction (LVEF). As such, HF can be either with preserved ejection fraction, HFpEF (LVEF > 50%), or reduced ejection fraction, HFrEF (LVEF < 40%), and patients with an LVEF between 40% and 50% are labeled as HF with mid-range EF-HFmrEF [3]. The European Society of Cardiology (ESC) highlights the fact that HFpEF diagnosis is challenging, emphasizing the current lack of consensus. According to ESC guidelines, there are four diagnostic criteria for HFpEF: the presence of HF symptoms and signs (dyspnea, orthopnea, cough), a LVEF of >50% (with the remark that patients with a LVEF 40%–49% may be classified as HFmrEF and included in clinical trials as HFpEF), increased levels of NPs (B-type natriuretic peptide: BNP > 35 pg/mL and/or N-terminal-Pro-BNP: NT-proBNP > 125 pg/mL) and imagistic evidence of structural heart disease, including LV hypertrophy, diastolic dysfunction and/or left atrial (LA) enlargement.

Biomarkers reflect myocardial damage and stress; systemic inflammation and fibrosis and their routine measurements mirror myocardial structural disease [2]. Natriuretic peptides are the most frequently used biomarkers in HF patients, as their elevated levels constitute a diagnostic criterion irrespective of LVEF [1,2,3,4]. However, the underlying mechanisms behind their increase differ in the two types of HF, with HFpEF patients showing lower levels of NPs [2,3,4,5]. In these patients, higher levels of circulating NP are given by the increased left ventricle (LV) diastolic filling pressure and end-diastolic wall stress [2,4]. As compared with HFrEF, the circulating levels of NPs are lower, especially in the context of lower myocardial stretch, LV end-diastolic wall stress and volume overload [4]. ESC guidelines recommend taking into consideration lower diagnostic thresholds for BNP and NT-proBNP when assessing a potential HFpEF patients [3].

Essential hypertension and myocardial ischemia are frequent causes of HFpEF and one third of these patients have concomitant atrial fibrillation (AF) [4]. The presence of AF impacts NPs circulating levels, with different studies showing that AF patients with HFpEF had higher mean NP levels compared to sinus rhythm patients [3,4,5,6,7]. As such, NP values must be interpreted differently in AF patients both when diagnosing HFpEF and assessing prognosis [2,4,7,8]. Several other factors affect NP levels [5,6,7,8,9]. Obese patients tend to have lower baseline NPs, while renal dysfunction, feminine gender and age are associated with increased levels [7].

## 3. Natriuretic Peptides: From Molecular Evidences to Clinical Implications

There are three endogenous NPs secreted as pre-prohormones: atrial natriuretic peptide (ANP), B-type natriuretic peptide (BNP) and C-type natriuretic peptide (CNP). Subsequently, there are three natriuretic peptide receptors (NPR): natriuretic peptide receptor A (NPR-A), natriuretic peptide receptor B (NPR-B) and natriuretic peptide receptor C (NPR-C or clearance receptor) [7,8,9,10,11,12,13,14,15,16]. These peptides act as hormones with pleiotropic effects by binding to the first two receptors, contributing to cardiovascular homeostasis and pressure and volume overload counter-regulatory mechanisms [12]. Moreover, all NPs have various molecular forms, which are different in healthy subjects from HF patients, such as different ANP forms and glycosylated proBNP [17].

Biologically active ANP is a 28 amino acid (aa) peptide. Initially secreted as pre-proANP (151 amino acids), this molecule is cleaved into proANP (126 amino-acids), which is deposited in granules inside atrial myocardium. A transmembrane protease cleaves the secreting proANP into its biologically active short-lived form (ANP- 28 amino acids) and its inactive form, NT-proANP (98 amino-acids) [7,8,9,10,11]. The latter has a much longer half-life (60–120 min); however, its numerous subsequent fragments make it a still unpractical biomarker in routine clinical practice.

The majority of ANP is secreted in the atria in response to myocardial stretch, atrial concentrations being 1000 higher as compared to ventricular ANP levels [17]. Ventricular myocardium produces small amounts of ANP; however, in HF patients, hypertrophied ventricular myocardium becomes able to secrete ANP [10]. Extracardiac sources include hypothalamus, lung and thyroid gland [5].

There are three forms of circulating ANP: αANP, βANP and pro-ANP [17]. Found only in the human atria, the biosynthesis pathways of the βANP are still unclear. Its structure is of an anti-parallel dimer of αANP. When compared to αANP, βANP has lower bioavailability (40%), slower onset of action and lower receptor affinity (Table 1).

ANP has numerous biological effects, including blood pressure lowering effects and renin-angiotensin-aldosterone system (RAAS) inhibition [16,17,18,19,20]. It may affect apical Na channel and Na/K ATPasis basal activity, determining a decreased sodium reabsorption and increased Na excretion, leading to increased natriuresis [20]. Moreover, ANP inhibits RAAS through renin (at the juxtaglomerular cyclic guanosine monophosphate- cGMP dependent cells level) and aldosterone inhibition [21], leading to blood pressure reduction. Moreover, studies have shown that ANP acts at the level of adrenal glands, directly inhibiting aldosterone production [16]. Not only does ANP inhibit RAAS, it also possesses antifibrotic and antihypertrophic effects through cGMP dependent angiotensin II and endothelin inhibition [12]. A study revealed that ANP might counteract myocardial hypertrophy by inhibiting calcium mediated epinephrine response through cGMP [15]. A different mechanism through which ANP affects blood pressure may be through baroreflex modulation, stimulating vagal afferent fibers [16].

Recent studies focusing on AF patients found that NT-proANP levels correlate with AF type and LA dimensions [18]. Seewoster et al. revealed that NT-proANP correlated with LA dimension as determined by cardiac magnetic resonance [18].

The gene coding ANP contains three exons: the first exon codes the 5′ region (not translated), a signal peptide formed of 25aa (16aa of proANP). The second exon is responsible for coding proANP while the third exon has a role in coding terminal 3′ tyrosine [5]. Recent studies have shown that a chorionic transmembrane enzyme cleaves proANP into pre-preoANP and NT-proANP [8].

The biologically active form of human BNP is BNP32 [8], widely varying across different species. The gene coding BNP is also formed of 3 exons: the first exon codes a 26 aa signal peptide and the first 15aa of proBNP; the second exon codes the majority of proBNP and the third exon codes terminal tyrosine and 3′ region [12]. mRNA BNP is translated into pre-proBNP with 134 aa after which the signal peptide is removed, resulting BNP-108 [1]. Different forms of BNP exist in the atria and the ventricles- BNP 32 and 108, respectively [12]. BNP32 is mostly found in the atria, while BNP 108 is found in the ventricular myocardial [21]. At the level of the Golgi apparatus, proBNP is cleaved into BNP and NT-proBNP, to be later released in plasma [21]. As opposed to ANP, which is mostly stored in vesicles, BNP is secreted in response to myocardial stretch.

Several studies identified higher BNP concentrations at the level of anterior interventricular vein and coronary sinus, which supports the fact that BNP is mainly secreted by ventricular myocardium [1]. The difference between BNP and ANP mRNA resides in a repetitive unit which determines mRNA BNP degradation in a fashion similar to oncogenes [4]. BNP gene expression differs from that of ANP, being more dynamic [9]. BNP possess vasodilator effects, promoting natriuresis and diuresis. At the myocardial level, BNP inhibits fibrosis and necrosis [16,17,18,19]. Also, it possesses anti-inflammatory effects through monocytes, B lymphocytes and natural killer cell regulation. Importantly, BNP interferes with post-skeletal muscles ischemia angiogenesis.

Importantly, it seems that HF patients have decreased amounts of BNP32; the latter is cleaved to either BNP3-32 or BNP8-32 by dipeptidyl peptidase IV (DPP IV). The two forms are less biologically active than BNP32, probably due to faster degradation and may account for the presumed resistance to NPs in HF patients [1,16].

Both ANP and BNP can undergo post-translational changes, such as phosphorylation and glycosylation. The physiological impact of a phosphorylated proANP form is still uncertain [17]. As one of the most frequent change, glycosylation stabilizes proteins, thus preventing further processing. The glycosylation pattern is strikingly different between ANP and BNP [17]. *O*-glycosylation of proBNP inside the Golgi apparatus is multi-sited (approximately seven sites) and its degree may vary among patients. Heavily glycosylated pro-BNP molecules are recognized as a cause of decreased conversion to its biologically active form, with a subsequent increase in pro-BNP/total BNP ratio. Interestingly, in acute HF the glycosylation level of proBNP decreases as there is a tendency towards forming more mature BNP. In acute HF, the increased proBNP production is accompanied by decreased glycosylation and increased furin activity, leading to elevated BNP and NT-proBNP levels.

C-type natriuretic peptide is secreted in the myocardium, endothelium, chondrocytes, brain and blood cells [12]. C-type natriuretic peptide gene also contains 3 exons, with the first exon containing 23 aa signal peptide and 7 aa proCNP. The second exon contains the proCNP sequence while the third the exon 3′ terminal. After removal from the 126 aa pre-proCNP of the 23 aa signal sequence results 103 aa proCNP. An intracellular endopeptidase cleaves proCNP into 53 aa CNP [16]. CNP_53_ is cleaved in return to CNP_22_. Both forms have the same functions; however, CNP_53_ is found in the myocardium while CNP_22_ is found in plasma and brain [21]. CNP also possess vasodilator effects, being secreted by endothelial cells in response to vascular lesions. Further, it inhibits fibrosis, platelet aggregation and tissue plasminogen activation. CNP levels tend to increase in advanced HF as compared to incipient HF.

NPR-A and NPR-B determine NP functions. ANP and BNP activate NPR-A while CNP activates NPR-B [10,11,12,13,14,15,16]. NPR-A is found in the lungs, kidneys, adrenal glands, while NPR-B is predominantly found inside fibroblasts [18,19,20].

NPR-C is found in the brain, atrium, lungs and aorta. These receptors have an extracellular region for the ligand attachment and a intracellular region with a cGMP dependent proteinkinase [6,22].

As a second messenger, cGMP is formed from two possible precursors, either soluble guanylyl cyclase- *sGC* (found in cytosol; it requires nitric oxide binding) and particulate guanylyl cyclase (pGC), found in the cellular membrane and activated via NPR. As such, this activation leads to an increase in cGMP, which in turn increases protein-kinase G (PKG) levels [22,23,24,25,26]. The latter phosphorylates several proteins, including myocardial cytoskeletal titin [26]. Moreover, decreased levels of cGMP and subsequently of PKG have been associated with myocardial remodeling through increased cardiomyocyte hypertrophy and resting tension [26] (Figure 1). HTN, AF, chronic kidney disease- frequently found comorbidities in HFPEF patients, determine a decrease in cGMP through a pro-inflammatory state and subsequent decrease of nitric oxide. Importantly, augmenting cGMP concentrations may constitute therapeutic targets in HFpEF.

NPs have a wide range of biological effects, including endocrine and paracrine. They promote diuresis and natriuresis, vasodilation and inhibit sympathetic nervous system and renin-angiotensin-angiotensinogen. In advanced HF there is a resistance to the effects of NPs, together with either an increased turnover, increased biologically inactive NP secretion or decreased NPR-A activation due to the dephosphorylation of secondary receptors [1,2,3,4,5].

If ANP levels are influenced by atrial pressures, BNP concentrations are determined by ventricular stretch in response to underlying pressure and/or volume overload. ANP’s short half-life (derived from its higher affinity for NPR-C) precludes its utilization in routine practice while BNP’s stability makes it a desirable biomarker in HF diagnosis and prognosis (Table 2).

A novel NP, middle-range proANP (MR-proANP) has emerged as having a greater stability and both diagnostic and prognostic values, especially in HFpEF patients. It derives from an intermediate region of NT-proANP and exhibits increased stability. Moreover, it correlates with increased LA dimensions and it seems that its diagnostic [24] and prognostic [12] utility might be superior to that of NT-proBNP in HFpEF. Moreover, it correlates with NYHA class in several studies [24].

The degradation of NPs can be either receptor- mediated (NPR-C) or enzyme-mediated [5,6,7,8,9,10] and is recognized as a therapeutic target in both hypertension and HF [12,16]. While receptor mediated NP degradation is based on internalization (claritin-mediated) and hydrolysis [16], enzyme-mediated NP degradation occurs mainly through the neutral endopeptidase (NEP) zinc-dependent neprylisin [4,10]. Although it is expressed by varying tissues, it can mostly be found in the proximal renal tubules, myocardium, fibroblasts and endothelial cells [4,5,6,7]. This enzymatic degradation makes both ANP and BNP unstable in serum, their plasmatic levels being used in routine clinical practice. Contrarily, NT-proBNP shows an increased serum stability. Inhibiting NEP leads to an increase in NP levels, which benefits HF patients [21]. Insulin degrading enzyme has also proven to degrade NPs, specifically ANP in addition to insulin [5].

## 4. Implications of Natriuretic Peptides in Heart Failure with Preserved Left Ventricular Ejection Fraction Diagnosis

Diagnosing HFpEF remains difficult due to a lack of consensus, patients’ heterogeneity and multiple concurrent pathologies that may mimic not only HF symptoms, but also lead to either increased (AF) or decreased (obesity) NP levels [1]. HFpEF diagnosis requires clinical and imagistic criteria, as well as elevated NPs levels. Given that one third of HFpEF have normal NP levels [1], relying solely on their values for diagnosis is not recommended and their value must be interpreted in the clinical context. As such, the diagnostic gold standard is cardiac catheterization showing increased LV filling pressures.

The longer plasma half-life of BNP and NT-proBNP (22 min and 70 min, respectively) as compared to ANP (2 min), makes the two the preferred NPs for guiding HF diagnosis and, probably, therapy. Moreover, despite having a longer half-life, NT-proANP failed to emerge as a diagnostic marker due to the increased number of cleaved fragments that limit its detection. European Society of Cardiology considers a BNP level of > 35 pg/mL and/or NT-proBNP > 125 pg/mL suggestive of chronic heart failure, with higher values being recommended in acute settings- over 100 pg/mL and > 300 pg/mL, respectively [1,3]. Moreover, it is agreed upon that acute HF patients exhibit increased NPs levels regardless of LVEF [3,27,28,29,30,31,32,33,34,35]. In a study, acute HFpEF patients showed NT-proBNP levels between 600–1000 pg/mL [36]. Although HFpEF patients/ patients previously treated with diuretics may exhibit lower levels, there is no consensus in what regards NPs diagnostic threshold. Given that NPs levels are affected by several factors, including the presence of AF and body mass index, their use rather resides in excluding the diagnosis of HF with a subsequent high-negative predictive value (0.94–0.98) [3]. As such, especially with respect to the diagnosis of HFpEF, NP levels must be corroborated with both the clinical context and other imagistic parameters. In the light of numerous affections that may mimic HFpEF symptoms, the ESC guidelines propose diagnostic thresholds of these NPs so as to limit over-diagnosing HFpEF.

Several studies agree that both BNP and NT-proBNP retain their diagnostic performance (although with a decrease in sensitivity and specificity) in acute heart failure, irrespective of LVEF [32]. However, factors that impact their levels should be taken into consideration when attempting HFpEF diagnosis. Both cardiac and non-cardiac causes may lead to a subsequent increase in NPs, including AF, recent cardioversion, myocarditis, acute coronary syndrome, age, severe renal impairment, pulmonary embolism, sepsis and critical illness [32,33]. It has been shown that the predictive values of NPs drop from 0.95 to 0.82 in patients > 75 years old [4]. As such, the expected NP values will be higher in the elderly, even in the absence of HF. Age-adjusted NT-proBNP have been proposed for acute HF diagnosis, considering cut-off values of >450 pg/mL, 900 pg/mL and >1800 pg/mL for patients <50 years, > 50 years and >75 years old, respectively [5]. In contrast, NPs levels tend to be lower in obese patients, irrespective of volume status [4]. The fact that NP concentrations vary among HFpEF patients is demonstrated by different medians across studies. The I-PRESERVE trial revealed a median NT-proBNP concentration of 341 pg/mL in HFpEF patients [34], while a different study highlighted that nearly a third of HFpEF patients had BNP levels below 100 pg/mL while displaying increased LV filling pressures as measured by cardiac catheterization [33].

Recent studies pointed out the need of different thresholds in HFpEF in regard to sinus rhythm patients. HFpEF and AF coexist in 30% of patients [37] while AF in itself is one of the most frequent causes of HFpEF. AF patients tend to have higher NT-proBNP levels and exercise intolerance in the absence of HF [1]. Moreover, they have increased LA dimensions and LV filling pressures, making echocardiographic findings of HFpEF difficult to interpret [1,37,38]. Although the use of higher levels is recommended, ESC has failed to provide a clear cut-off value for HFpEF diagnosis in AF patients. Several studies reported the use of different NPs cut-off values as inclusion criteria with respect to underlying cardiac rhythm (sinus versus atrial fibrillation): 600 pg/mL in the SOCRATES trial [39] and >900 pg/mL in PARAGON trial [40].

The use of different diagnostic methods in adjunction with increased NPs level is recommended by ESC guidelines. Structural and functional alterations as determined by transthoracic echocardiography include a left atrial indexed volume (LAVI) of >34 mL, increased LV mass index (115g/m^2^ for men and 95 g/m^2^ for females), E/e’ >13 and mean septal and lateral wall e’ of <9 cm/s [3]. This is supported by the correlation between BNP and structural and functional alterations in HFpEF. In a study conducted by Iwanaga et al., BNP levels correlated with both left ventricular end-diastolic pressure and end-diastolic wall stress, more significantly with the latter [41]. Moreover, it seems that a BNP of > 100 pg/mL or a NT-proBNP of > 600 pg/mL indicates a LV restrictive filling pattern [32]. NP levels also correlate with LA dimensions, this correlation being stronger in HFpEF patients [35]. Although LVEF is preserved, it seems that global systolic function is altered in HFpEF patients. In a study conducted by Kraigher-Krainer et al., the decreased LV systolic strain (both longitudinal and circumferential) noticed in HFpEF patients correlated with NT-proBNP levels [38]. The disposition of LV hypertrophy also influences BNP levels, being significantly elevated in patients with concentric as compared to eccentric LV hypertrophy [5].

In the setting of an acute coronary syndrome, ANP levels show an early elevation with a rapid decline, as opposed to BNP levels, who tend to exhibit a bimodal elevation [5]. The first peak has been reported in the first two days post-myocardial infarction, with the second occurring nearly one week after the event, reflecting the extent of LV remodeling.

Not only that the diagnostic performance remains constant, but the treatment is similar in decompensated HF with regard to LVEF [3,40,41,42,43,44,45,46,47]. In chronic HF however, the treatment differs significantly between HFpEF and HFrEF, being well-established for the latter.

Renal function is tightly related to NPs levels. NPs tend to be elevated in CKD and end-stage renal disease, up to a value of 200 pg/mL even in the absence of overt HF [5]. Haemodialysis but not peritoneal dialysis has been shown to lower BNP levels with nearly 40% [43]. Proposed cut-off values of NT-proBNP for diagnosing HF in CKD patients include a level of > 1200 pg/mL, with higher levels being suggested in older patients [5,44].

Recently, a new NP has emerged as a potential HFpEF diagnostic tool. Cui et al. have revealed increased MR-proANP levels in HFpEF patients, with a significantly higher AUC when compared to NT-proBNP (0.844 versus 0.518, *p* <0.001) [24]. Moreover, when comparing their levels based on NYHA class, MR-proANP concentrations differ in regard with NYHA class, as compared to NT-proBNP, which showed no variation. Taking into consideration the link with echocardiographic parameters, MR-proANP correlated with LAVI, as opposed to NT-proBNP. In another study, MR-proANP showed non-inferiority to NT-proBNP in acute HF diagnosis, being elevated even in patients who showed non-diagnostic NT-proBNP levels [27]. The authors of the BACH study found that a MR-proANP of >120 pmol/L was suggestive of HF; adding this parameter to BNP increased its diagnostic performance to 73.6% [27].

Not only have NPs proven their diagnostic utility, several studies have questioned their ability to identify patients at risk for HF development. The STOP-HF trial referred patients with BNP levels of > 50 pg/mL to further echocardiographic investigations, leading to a decrease in LV dysfunction [42].

Given its difficulties, the diagnosis of HFpEF in its characteristically heterogeneous population usually requires more than one biomarker. As the separation of HF in reduced and preserved EF occurred rather recently, more studies focused on HFrEF. Although both European and American guidelines have yet to consider different thresholds for the two classes of HF, increasing evidence links their distinct pathophysiologies to unique therapeutic strategies. Moreover, taking into consideration the increasing HFpEF incidence and altered prognosis, proper identification of these patients become vital.

## 5. Therapeutic Implications of Natriuretic Peptides in Heart Failure with Preserved Left Ventricular Ejection Fraction

NPs have been shown to inhibit RAAS, suppressing angiotensin II mediated vasoconstriction, sodium reabsorption (proximal tubule) and aldosterone, endothelin and renin secretion [5,9]. Their use in HF therapy is bimodal, both as a therapeutic target per se and as an indicator evaluating therapy response. However, their use in HFpEF remains controversial.

The rationale behind using NPs as a therapeutic target in HF therapy [45,46,47,48,49,50,51,52,53,54,55,56,57,58,59,60,61,62] resides in the seemingly abnormal BNP processing with a subsequent deficiency in active forms and resistance to their biological effects in these patients [5]. It seems that HF patients are deficient in biologically active BNP32 with a subsequent increase in BNP1–108 [45]. Augmenting their effects can be done either by administering NPs or reducing their breakdown.

### 5.1. Natriuretic Peptides Analogs

The use of recombinant NPs in acute HF is recommended by ESC [3], but without any distinction regarding EF. Nesiritide is a recombinant form of BNP used as a vasodilator in acute HF which increases GMP levels by binding to NPR-A [3,5]. Apart from its vasodilator effects, is has shown to inhibit apoptosis and limit myocardial remodeling and subsequent hypertrophy [5,50]. ESC guidelines recommend its use especially in hypertensive acute HF, very frequently associated with a preserved LVEF. However, patients with acute HFpEF tend to respond differently to vasodilators than their counterparts with reduced EF. While BP reduction tends to be greater in response to vasodilators, improvement in stroke volume is minimal [47]. The ASCEND trial showed that nesiritide leads to minimal clinical improvement in HFpEF patients, with no impact on renal function, hospitalization rates or death [42]. A different study showed that nesiritide failed to impact both symptoms and clinical outcomes, with lower decreased sodium excretion and subsequent weight reduction and increased rates of therapy failure [49]. Despite not being formally contraindicated and while its current use as a vasodilator is a class IIaB indication, the role of nesiritide in acute HFpEF therapy remains controversial with further studies required to assess its benefits.

A different NP used in HF treatment is carperitide, a recombinant form of ANP. Although recommended in Japan for acute HF therapy, it has failed to enter either ESC or American guidelines, especially because of limited studies and decreased half-life [5,51]. Despite improving symptoms in the studied Japanese population, there was no impact reported on mortality. Furthermore, the impact on HFpEF patients is unknown. Urodilatin is a form of ANP which determined clinical improvement in acute HF patients, but at the cost of worsening renal function and hypotension [5]. Its routine use in acute HF therapy has yet to be approved due to the lack of impact on mortality, hospitalization rates and myocardial injury [52]. Its use in HFpEF has yet to been studied.

Attempts have been made to produce variants by the means of native NP gene alteration, resulting in NPR-A binding peptides resistant to degradation with subsequent increased half-life [1,5]. Several designer NPs (Table 3) have already been produced, seemingly retaining the anti-fibrotic, anti-hypertrophic and vasodilator properties of the native NPs: cenderitide-NP [53], CU-NP [56], M-ANP [58], AS-BNP [59]. Although they haven’t been approved for HF therapy, some have entered phase II trials (CD-NP).

### 5.2. Natriuretic Peptide Breakdown Inhibitors

Another possible therapeutic strategy is inhibiting the breakdown of NPs, thus promoting their biological effects [63,64,65,66,67,68,69,70,71,72,73,74,75,76,77,78,79,80,81]. This can be done by inhibiting NEP, a zinc dependent enzyme that represents the final step in NPs cleavage. NEP has a large distribution, being found inside cardiomyocites, fibroblasts, central nervous system (brain, cerebrospinal fluid), immune cells (neutrophils), lungs, endocrine glands (thyroid, adrenal glands), myocardium. Its substrate is as extended as its distribution, acting on angiotensin I, II, bradykinin, endothelin-1, glucagon and amyloid-beta [5,81].

There are several drugs that inhibit NEP (Table 4). Pure NEP inhibitors, such as candoxatril [62] and ecadotril [5,62] have been shown to raise NPs, but their ability to stimulate RAAS at the same time together with uncertain and contradictory effects on blood pressure, clinical improvement and survival precluded their entry in routine practice [81].

A different strategy that has been attempted targets dual endothelin and NP breakdown inhibition. It seems that endothelin-1 (ET-1) and angiotensin II share common pathophysiological mechanisms. Although ET-1 impacts HF progression through vasoconstrictor and fibrotic effects, antagonizing its receptors have failed to show prognostic effects in HF patients [5]. However, inhibiting endothelin-converting enzyme (ECE) and NP cleavage led to an improvement of symptoms through RAAS inhibition, with a subsequent decrease in ET-1 levels apart from an increase NPs (dual ECE/NEP inhibitors). Moreover, it seems that ECE also plays a role in NP degradation, as its structure chemically resembles that of NEP [65]. Different drugs have been created [64,67], however none implemented yet due to the lack of clinical trials. Drugs capable of inhibiting ECE, NEP and ACE have been produced; however there are serious concerns about their safety [71].

Sacubitril/valsartan (formerly known as LCZ 696) combines a NEP inhibitor (sacubiril) with an angiotensin receptor blocker (valsartan) in a 1:1 ratio, being the first in its class (ARNI). Its benefits come from combining neprilysin inhibition, which limits NPs breakdown and leads to a subsequent rise in both their levels, and cGMPs with RAAS inhibition given by valsartan [5,74]. Among the NPs, both ANP and BNP are substrates for neprilysin; however, its affinity is stronger for the former [1,9,74,75,76,77,78,79,80,81,82,83,84,85,86]. Subsequently, the increase in ANP levels (and subsequently in cGMP) will surpass that of BNP. However, due to its short plasma half-life, BNP is preferred over ANP for monitoring its effects. Of note, NT-proBNP is not a direct substrate for neprylisin inhibition; as such the decrease in its levels makes it the preferred biomarker for monitoring HF patients treated with sacubitril/valsartan, especially during the first 10 weeks, when the increase in BNP levels is maximum [74,75,76,87]. Compared with dual ACE/NEP inhibitors, the incidence of angioedema is lower due to a lower increase in bradykinin [74,75,76,77].

Several studies support the sacubitril/valsartan treatment class IB indication in HFrEF patients [3,73,74,75,76,88]. However, there is no clear indication for this drug in HFpEF and there are no clinical trials assessing outcomes in patients treated with the ARNI. Moreover, a Swedish study revealed an undiagnosed/lower NP level as the most frequent cause of HFrEF patients not receiving sacubitril/valsartan [72]. This could easily apply in HFpEF, given the controversy of NP levels in this set of patients [5,74].

Regarding cGMP and sacubitril/valsartan therapy, there is extensive focus on its beneficial effects on cGMP levels and, subsequently, PKG. cGMP deficiency is of paramount importance in HFpEF, as it lowers PKG levels and thus promotes myocardial remodeling (both hypertrophy and impaired relaxation through increased cardiomyocyte resting tension) [26]. Through oxidative stress [21] and a subsequent decrease in NO availability [26], numerous comorbidities contribute to the lower levels of cGMP found in HFpEF. It seems that the pro-inflammatory state induced by these comorbidities triggers coronary microvascular inflammation, which further decreases NO availability [26] and inhibits cGMP formation. Moreover, oxidative stress affects titin, a protein found in cardiomyocytes cytoskeleton in two ways: firstly by inhibiting PKG- dependent phosphorylation and secondly by determining the formation of disulfide bridges, which leads to a more stiffer titin molecule [26]. The authors agree that the cGMP deficiency in HFpEF can be ascribed to decreased production and not increased breakdown [25], which explains why in HFpEF patients, cGMP levels increase in response to neprylisin inhibitors and not phosphodiesterase inhibitors, which inhibit cGMP breakdown [21].

The PARAMOUNT trial assessed the benefits and safety profile of sacubitril/valsartan treatment in HFpEF patients [84]. NT-proBNP levels significantly decreased in sacubitril/valsartan group, especially in diabetic patients. After 36 weeks of treatment, there was an improvement in NYHA class and a reverse LA remodelling with a subsequent decrease in LA volume (especially in sinus rhythm patients), and no changes in LVEF or LV volumes.

Recently, the baseline characteristics of patients enrolled in PARAGON-HF trial, which assessed the impact of sacubitril/valsartan therapy in HFpEF on mortality, have been published [85]. 4822 patients with HF NYHA class II-IV, elevated NPs, LVEF >45% and increased LA dimensions/ LV hypertrophy were included. Regarding NPs levels, the median NT-proBNP was 885 pg/mL, with higher thresholds being used for AF and previously HF hospitalized patients’ inclusion. In the absence of recent (<9 months) HF hospitalization, NT-proBNP inclusion concentration thresholds used were >300pg/mL for sinus rhythm patients and >900 pg/mL for patients, the levels being lower in recent hospitalized patients (>200 pg/mL and >600 pg/mL, respectively). Again, the most common cause of exclusion was lower NPs levels. The final results will show whether combined ARB/NEP inhibition could be a therapeutic option in HF patients with an LVEF >45%.

An interesting possible side-effect of sacubitril/valsartan therapy may be an increase in amyloid-*β* concentration as NEP is responsible with enzymatic clearance of amyloid, thus augmenting the risk of developing Alzheimer’s. An ongoing trial will assess the impact of the novel drug as compared to valsartan treated HFpEF patients on cognitive impairment, with results being scheduled for 2022 [79].

### 5.3. Mineralocorticoid Receptors Antagonists

Apart from novel therapies, several studies investigated the benefits of mineralocorticoid receptors antagonists (MRAs) therapy in HFpEF [6,89,90,91,92]. The sub analysis of the TOPCAT trial conducted by Anand et al. reveals that while spironolactone does not impact overall mortality and hospitalization rates, it proved beneficial in HFpEF patients with lower NPs levels and not in those with increased concentrations [88]. These results were similar to those of the I-PRESERVE trial [34], in which Irbesartan treatment was beneficial in HFpEF patients with lower NPs levels. The results of the two trials point out the fact that higher NPs levels in HFpEF translate an increased cardiovascular event rate, not necessarily increased treatment responsiveness. Moreover, it seems that higher NP levels in these patients relate to a more advanced degree of structural heart disease, including higher degree of myocardial fibrosis, less responsive to therapeutic intervention [88].

Patients with HFpEF present increased fibrosis biomarkers, as shown by Zile et al. [89] and Oikonomou et al. [6] In a study conducted by Cho-Kai Wu et al. [20], HFpEF patients with increased myocardial fibrosis degree as assessed by late-gadolinium enhancement magnetic resonance imaging scans had higher levels of NT-proBNP. Fibrosis biomarker concentrations (growth differentiation factor, galectin-3, tissue inhibitor of metalloproteinase, matrix metalloproteinase 2) correlated with the severity of fibrosis. However, the authors concluded that NPs rather reflect wall tension and volume overload with lower discriminative power in regard to the presence and/or degree of myocardial fibrosis.

It appears that HFpEF with lower matrix metalloproteinase 9 levels benefit most from eplerenone treatment [6]. Chen et al. [90] show that MRA therapy in HFpEF reduce both myocardial remodeling and amino terminal peptide of procollagen III levels, without however affecting NP levels. It may be that in the early stages of HFpEF, the progression of myocardial remodeling and subsequent fibrosis is preventable by therapeutic interventions, such as MRAs therapy, thus explaining the benefit that patients with lower fibrosis biomarkers and NPs levels have from this therapy.

## 6. Prognostic Implications of Natriuretic Peptides in Heart Failure with Preserved Left Ventricular Ejection Fraction

It is agreed upon that although NP values tend to be lower in HFpEF patients with no consensus regarding diagnostic thresholds, they retain their prognostic utility irrespective of LVEF [3,87].

In a different study, increased BNP levels were associated with increased mortality, irrespective of LVEF [93]. Also, it seems that higher BNP levels translate into increased risk for developing both AF and transient ischemic attacks [1].

### 6.1. Prognostic Value of BNP and NT-proBNP

The NPs concentrations retain their prognostic utility albeit their lower baseline levels in HFpEF patients [94]. Several studies reported similar death risks for a given NPs concentration irrespective of HF phenotype [93,95,96,97,98,99]. Levy and Anand compared patients from I-PRESERVE and VALHEFT trials, showing that a 1-log increase in NT-proBNP levels carries a mortality HR of 1.7 regardless of LVEF [96]. However, the same authors demonstrated that the mortality of HFrEF patients was two thirds higher than of those with HFpEF. Similarly, Salah et al. reported that hospitalized HFpEF patients have lower mortality rates when compared to HFrEF [98]; however, the difference is minor and the risk of death tends to equalize after discharge. More importantly, although patients with HFpEF display lower baseline NPs concentrations, for the same NT-proBNP level, prognosis is similar regardless of LVEF. This paradox can be firstly be explained by the different mechanisms involved in NPs secretion between the two phenotypes. It is known that at the same end-diastolic LV pressure, HFpEF patients display lower NT-proBNP levels. This may be due to the correlation of NPs with diastolic wall stress, which, according to the law of La Place, correlates with wall pressure and cavity diameter and is inversely related to wall thickness. As patients with HFpEF typically exhibit a concentric LV remodelling with increased wall thickness, this may explain in part their lower NPs levels for a given increased wedge pressure as compared to the HFrEF patients (who typically have an eccentrical LV remodelling). Secondly, this paradox can be explained by the distribution of comorbidities in regard with HF phenotype. As such, HFpEF patients tend to be older, with increased incidence of arterial hypertension, chronic kidney disease, AF and anemia, comorbidities that may account for similar prognosis between HF phenotypes despite lower NTpro-BNP of HFpEF [95,98]. This drives attention to the need of also addressing non-cardiovascular diseases in order to improve outcomes in HFpEF patients. The fact that while the same NP concentration may point out to the same relative risk of death and overall prognosis still differs between HF phenotypes underlines exactly the contribution of other risk factors and comorbidities to the mortality. That’s why it is not advisable to regard NP levels as surrogate markers of mortality.

It appears that different NPs concentrations threshold might be necessary for prognosis in HFpEF patients as compared to HFrEF [94,95,96,97,98,99,100,101,102,103,104,105,106]. Accordingly, NPs prognostic values vary across different studies [34,87,88], several authors highlighting that NPs increase in addition to their baseline levels holds prognostic importance [34]. Moreover, relying on an NP prognosis threshold may not be necessary, as prognosis can and should be continuously assessed.

Anand et al. associated a baseline NT-proBNP of 339 pg/mL with a four year mortality of 21.1%, which translates into a 5% annual mortality for an NT-proBNP level between 300–500 pg/mL [34].

As NT-proBNP is also influenced by AF [34], increasing its concentration irrespective of HF presence [37], these levels should be judged accordingly when determining prognosis. Kristensen et al. showed that different NT-proBNP levels should be used in AF versus non-AF patients to determine prognosis. While an NT-proBNP level of < 400 pg/mL was associated with a better prognosis irrespective of rhythm; the presence of AF accounted for the different mortality rates in HFpEF patients with a NT-proBNP > 400 pg/mL [107]. However, higher NT-proBNP AF patients had increased risks of hospitalization as compared to patients with HFrEF and the same NPs level. In addition, it seems that patients with lower baseline NPs levels would benefit on long-term from supplementary explorations [87].

A recent study launched the possibility of increased NT-proBNP playing a causative role in AF [100]. This idea is controversial as several comorbidities that promote AF lead in turn to an increase in NT-proBNP, which remains a biologically inactive product. Moreover, an increase in both ANP and BNP in sacubitril/valsartan patients led to atrial reverse-remodeling, emphasizing their antifibrotic and antihypertrophic effects [87].

### 6.2. Prognostic Value of MR-proANP

Another emerging prognostic biomarker is MR-proANP [27]. It seems that MR-proANP accurately assesses mortality in both acute and chronic HF [27,30]. It seems that increased MR-proANP levels correlate with an increased four year mortality risk [106]. Moreover, in a Swedish study, the novel biomarker predicted both HF and AF, as opposed to NT-proBNP which failed to predict AF [97]. The study assessed the difference in AF versus non-AF HFpEF patients’ prognostics as determined by their NT-proBNP level. Alehagen et al. also found MR-proANP to be prognostic of cardiovascular mortality at five years for HFpEF patients [108]. Similarly, Zabarovskaja and colleagues stated that a MR-proANP of >313 pmol/L predicted all-cause mortality (Table 5) [109]. The association between NP and AF has been studied [18,109,110,111,112,113]. Although ANP levels seem to decrease after electrical cardioversion [112] and are associated with AF progression [18], both BNP and NT-proBNP are preferred to assess AF recurrence rates post electrical cardioversion [114].

### 6.3. Prognostic Value of Natriuretic Peptides and Heart Failure with Preserved Ejection Fraction Therapy

There are controversies regarding the prognostic value of NPs in sacubitril/valsartan treated patients. As the formerly known LCZ696 interacts with NPs levels, there still are questions regarding their accuracy in determining HF patients’ prognosis. Some studies agree that both BNP and NT-proBNP retain their prognostic utility in these patients, emphasizing the fact that during therapy initiation, NT-proBNP is the preferred biomarker (several months are required for BNP levels stabilization) [74]. However, the currently available trials did not include HFpEF patients treated with sacubitril/valsartan.

Another issue regarding NPs prognostic value in HFpEF is whether these patients would benefit from NPs guided therapy. So far, studies are controversial [3,46,115,116,117,118,119,120,121,122,123,124,125,126,127], especially taking into consideration that these patients respond differently to HF therapy than those with reduced LVEF. A study conducted by Khan et al. including patients with both HFrEF and HFpEF came to the conclusion that NP guided therapy was not beneficial [115]. A different study emphasized that regardless of LVEF, a constantly elevated NPs should not determine changes in patient management [117]. Maeder et al. revealed in TIME-CHF trial that NT-proBNP guided therapy was not beneficial in HFpEF patients, as compared to HFrEF [118]. Although it included HFrEF patients, the GUIDE-IT trial concluded that biomarker directed therapy may not benefit these patients [122]. The opposite was shown in a meta-analysis conducted by Troughton et al. [119]. The authors stated that BNP-guided therapy improved outcomes in patients < 75 years old and reduced HF hospitalization rates regardless of age and LVEF. Brunner la Roca et al. agreed that NP guided therapy either through BNP or NT-proBNP is safe and more importantly, cost-effective. [4]. A British team stated that BNP-guided therapy might be cost-effective for HFpEF patients < 75 years old [120].

Recent studies have focused on MR-proANP as a possible biomarker to guide HF therapy. So far, it seems that MR-proANP levels might be able to predict incident HF, as it shown in a study conducted by Sabatine et al. [124]. Interestingly, it seems that MR-proANP might predict cardiac resynchronization therapy responders which showed lower levels of the biomarker as compared to the non-responders [125].

Regardless of the biomarker used, NP guided medical therapy remains controversial, especially in HFpEF patients. In these patients, more studies are required to assess the benefits of optimizing medical therapy after serial NP measurements. The most studied NPs are BNP and its inactive form, NT-proBNP, especially due to their proven diagnostic capacities. The novel MR-proANP, with its increased half time as compared to ANP, shows promise both in prognosis and guiding medical therapy, but more studies are required to confirm its safety profile and utility.

## 7. Future Perspectives

Several on-going studies on NPs in HFpEF have been announced. The results of PARAGON-HF trial will clear the impact of sacubitril/valsartan therapy in HFpEF [85], while another on-going study conducted by Mayo clinic (ClinicalTrials.gov Identifier: NCT03506412) will determine how this therapy affects NPs levels and cGMP in HFpEF patients [127]. A different study finishing in December 2019 will assess the effects of sacubitril/valsartan as compared to either enalapril or valsartan in HFpEF patients (ClinicalTrials.gov Identifier: NCT03066804) [128].

Moreover, in the context of the pro-inflammatory state of HFpEF, there is increasing body of evidence linking non-NPs biomarkers to HFpEF as a better diagnostic tool [21]. Emerging markers reflecting myocardial fibrosis such as a soluble source of tumorigenicity 2 (sST2), growth differentiation factor-15 [21], galectin 3 [126] and interleukins (1 and 6) are showing promise, correlating in different degrees with transthoracic echocardiographic parameters of diastolic dysfunction [10,21]. Their utility in comparison to NPs remains to be evaluated.

## 8. Conclusions

The number of HFpEF patients is increasing; moreover, these patients show similar prognosis with HFrEF patients.

The role of NPs in HF is both diagnostic and prognostic. Guidelines regard a BNP > 35 pg/mL or an NT-proBNP > 125 pg/mL as being diagnostic in non-acute setting, while emphasizing the fact that although several factors alter NP concentrations, their diagnostic utility rather lies in their negative predictive ability. For decompensated HF, either a BNP > 100 pg/mL, an NT-proBNP > 300 pg/mL or a MR-proANP of > 120 pmol/L is diagnostic. Authors agree that lower levels may be considered for HFpEF, without any thresholds being agreed upon. The absence of a consensus regarding diagnostic thresholds for NPs in HFpEF has led to inhomogeneous inclusion criteria in large clinical trials, thus affecting the results.

The role of NPs extends beyond diagnosis, as they can be regarded as a therapeutic target per se through nesiritide and the novel ARNI, sacubitril/valsartan. Augmenting the concentrations of NPs leads to lower blood pressure levels as well as an improvement in symptoms and quality of life. A degree of reverse remodelling with a subsequent decrease in myocardial fibrosis and LA dimensions could be determined.

Higher NPs levels are associated with poorer prognosis, especially in acute settings, irrespective of LVEF. Moreover, BNP and MR-proANP predict not only incident HF, but also AF. As AF is one of the most frequent causes of HFpEF, this becomes of the utmost importance. For AF patients it must be remembered that NP levels tend to be higher, irrespective of the presence of HF. Also, for the same BNP concentration, they show a better prognostic as compared with their sinus-rhythm counterparts.

Finally, guiding medical therapy by serial NP measurements is controversial, especially in HFpEF. Authors recommend in the absence of consensus that no change in patient management should be taken if the NP levels remain constantly elevated throughout treatment.

The need for further studies assessing diagnostic NP thresholds and prognostic values in HFpEF patients is enormous. Furthermore, given the novel therapies that interact with NPs, such as sacubitril/valsartan, these levels might be altered.

## Figures and Tables

**Figure 1 ijms-20-02629-f001:**
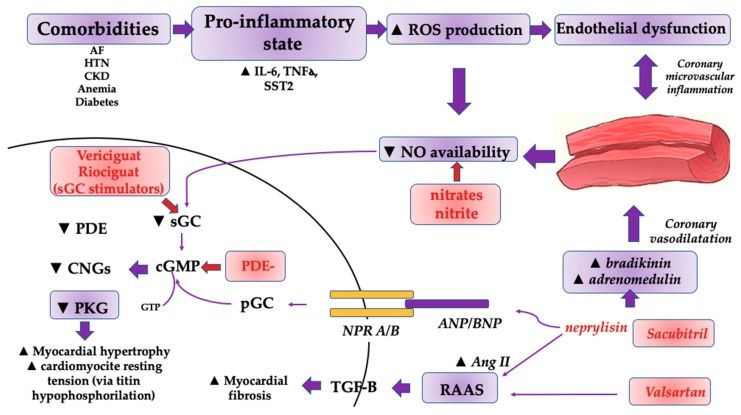
NPs, cGMP and RAAS in HFpEF patients and possible therapeutic targets. AF: atrial fibrillation; Ang II: angiotensin II; cGMP: cyclic guanosine monophosphate; CKD: chronic kidney disease; CNGs: cyclic nucleotide gated-ion channels; HTN: hypertension; IL: interleukin; NO: nitric oxide; NP: natriuretic peptide; NPR: natriuretic peptide receptor; pGC: particulate guanylyl cyclase; PKG: protein kinase G; PDE: phosphodiesterase; PDE-: phosphodiesterase inhibitors; RAAS: renin-angiotensin-aldosterone system; sGC: soluble guanylyl cyclase, TGF-B: transforming growth factor beta.

**Table 1 ijms-20-02629-t001:** Various forms of atrial natriuretic peptide. ANP- atrial natriuretic peptide; LV- left ventricle.

ANP Form	Structure	Effects	Additional Remarks	Reference
αANP	Compact ring structure	Prolonged bioavailability as compared to other ANP forms	Healthy subjects: αANP>>proANP>>βANP Heart failure patients: Increased βANP and pro-ANP concentrations with decreased circulating corin levels Increased βANP/total ANP correlates with LV dysfunction	[17]
βANP	Flexible extended structure αANP antiparallel homodimer	40% of ANP effects		
proANP	precursor	10% of ANP effects (weak natriuretic)		

**Table 2 ijms-20-02629-t002:** The biological characteristics of natriuretic peptides. ANP- atria natriuretic peptide; BNP- B type natriuretic peptide; CNP- C-type natriuretic peptide; * values may vary slightly across studies.

Natriuretic Peptide	Mechanism	Time	Normal Levels	Reference
BNP	Ventricular wall stretch (pressure/volume overload)	20 min	3.5 pg/mL	[4,5,6,7,8,9,10,11,12,13,14,15,16,17]
NT-proBNP	Biologically inactive form of BNP	60–90 min	51 pg/mL	
ANP	Atrial wall stretch (pressure/volume overload)	2 min	20 pg/mL	
NT-proANP	Biologically inactive form of ANP	60–120 min	0.11–0.60 nmol/L	
CNP	Endothelial lesions	2.6 min	Nearly undetectable	

**Table 3 ijms-20-02629-t003:** Designer natriuretic peptides as potential therapies in heart failure.

Designer Natriuretic Peptide	Structure	Effects	Reference
CD-NP (Cenderitide)	Fusion between CNP-22 and 15 aa DNP C-terminal	Vasodilator Antifibrotic, antiproliferative ↑GFR, ↓LA pressure (via NPR-A, NPR-B) less hypotension than nesiritide (minimal changes in BP)	[53,54,55]
CU-NP (humanized version of cenderitide)	Fusion between 17 aa ring of CNP and urodilatin’s N-terminal	+ cGMP => RAAS inhibition antihypertrophic (NHE-1/calcineurin pathway inhibition) renal function enhancement	[56,57]
M-ANP (Mutant-ANP)	12 aa extension to native ANP’s C-terminal	Enhances natriuresis and diuresis RAAS and sympathetic nervous system inhibition Inhibits cellular proliferation Antifibrotic	[58]
ANX-042 (AS-BNP)	Fusion between AS-BNP’s 16 aa of C terminal and 26 aa of native BNP	+ cGMP; RAAS inhibition natriuresis and diuresis stimulation	[57,59,60]
Nesiritide	Recombinant BNP	Vasodilator Hypotension Less renal function deterioration as compared to diuretic treated HFpEF patients	[61]
Carperitide	Recombinant ANP	Vasodilator Renoprotective Not widely recommended	[51]

ANP- atrial natriuretic peptide; AS-BNP- alternatively spliced BNP; aa-amino acids; BNP- b type natriuretic peptide; CNP- c type natriuretic peptide; cGMP- cyclic guanosine monophosphate; GFR- glomerular filtration rate; LA- left atrium; M-ANP: mutant ANP; NP- natriuretic peptide; NPR- natriuretic peptide receptor; NHE- natrium-proton exchanger; RAAS- renin-angiotensin-aldosterone system.

**Table 4 ijms-20-02629-t004:** Dual and triple endothelin converting enzyme, neutral endopeptidase and angiotensin converting enzyme inhibitors.

Class	Drug	Effects	References
Pure NEP inhibitors	Candoxatril	↑NPs and natriuresis ↑angiotensin II (RAAS stimulation) => vasoconstriction	[62]
	Ecadotril	No proven clinical benefit; may determine aplastic anemia	[63]
Dual ECE/NEP inhibitors	Daglutril (SLV-306)	↑NPs, ET-1, ↓BP, LV hypertrophy and pressure	[67]
	SLV-338	↓ LV remodelling (independently of BP lowering effects)	[64]
Dual NEP/ACE inhibition	Sampatrilat	Despite clinical benefits, its short half-life precluded its clinical use	[68]
	Omapatrilat	Symptoms relief and improved survival; ↓BP, vascular resistance (more than candoxatril) LV remodelling, myocardial fibrosis hypotension and angioedema (bradykinin)	[68,69,70]
Triple ACE/ECE/NEP inhibitors	Benazepril (ACE) + CGS 26303 (dual ECE/NEP inhibitor)	↑NPs, bradykinin inhibits angiotensin II and ET-1 ↓ LV remodelling (including mass and end-diastolic pressure)	[71]

ACE- angiotensin converting enzyme; BP- blood pressure; ET-1- endothelin- 1; ECE- endothelin converting enzyme; NEP- neutral endopeptidase; NPs- natriuretic peptides, LV- left ventricle.

**Table 5 ijms-20-02629-t005:** Prognostic values of natriuretic peptides in heart failure with preserved ejection fraction patients.

Biomarker	Concentration	Utility	Reference	Additional Remarks
BNP	>540 pg/mL	Predicts in-hospital mortality	[93]	May be able to predict AF Levels affected by sacubitril/valsartan
NT-proBNP	>300–500 pg/mL (339 pg/mL)	5% 1 year mortality	[34]	AF patients hold better prognosis at the same NT-proBNP levels as their sinus rhythm counterparts
MR-proANP	>313 pmol/L	Predicts all-cause mortality	[109]	May be able to predict AF; correlates with LAVI

BNP- B-type natriuretic peptide; MR-proANP- middle region pro atrial natriuretic peptide; AF- atrial fibrillation; LAVI- indexed left atrial volume.

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
