# Peer review of "Natriuretic Peptides in Heart Failure with Preserved Left Ventricular Ejection Fraction: From Molecular Evidences to Clinical Implications"

_ijms, 2019, doi:10.3390/ijms20112629_

Round 1
Reviewer 1 Report
The review on "Implications of natriuretic peptides in heart failure with preserved ejection fraction: from molecular evidences to clinical implications" summarizes what is known about natriuretic peptides as diagnostic, prognostic and therapeutic instruments, in the subset of heart failure patients with preserved ejection fraction.
The review seems to have originated from a question of diagnostic uncertainty (what are the proper cut-off levels in natriuretic peptides for diagnosing HFpEF), but also therapeutic uncertainty as it addresses the absence of therapies in HFpEF, where the sacubitril/valsartan RCT trial PARAMOUNT still has to be published. The title mentions 'implications' twofold: "Implications of natriuretic peptides" and "from molecular evidences to clinical implications". So I would suggest: "Natriuretic peptides in heart failure with preserved left ventricular ejection fraction: from molecular findings to clinical implications".
I can accept part of the paragraphs 3 (molecular evidence), and 5 (therapies) although I would advise 1) (paragraph 3) to make a figure of the NP effects on downstream molecular mechanisms (RAAS, cGMP) via the receptors, and 2) (paragraph 5) to incorporate the BNP analogs nesiritide and carperitide in Table 2. Please make a proper distinction in this table in the effects that are observed specifically in HFpEF patients; if it was only used for HFrEF patients, then please only mention shortly, because this is not for the present review on HFpEF.
At this time, I can only report the larger comments, and in general, the text is too extensive; it may be shortened with ease, if you can focus on the main messages. The text may be interesting for the reader after major revision.
Main Comments are on Paragraphs 1, 2, 6.
There are some viewpoints that to me are incorrect and have to be improved:
1. Abstract line 34 and introduction line 52: HFpEF "patients share the same prognosis as those with reduced ejection fraction" . There are no references added. Although this was demonstrated in CHF patients in the past (Varela-Roman et al, Heart 2005;91:489–494), since the MAGGIC publication (Eur Heart J 2012;33:1750–7), and the publication of Lam et al (Eur Heart J 2018;39:1770–80), this is no longer a definite fact: chronic HFpEF patients may have a better prognosis than HFrEF patients. Still, the HR's for mortality of natriuretic peptide levels are the same for HFpEF and HFrEF patients, as can be seen from comparing I-PRESERVE and VALHEFT studies (Levy and Anand, JACC HEART FAILURE 2 0 1 4 : 4 3 7 – 9). This is alos true for patients hospitalised with heart failure: (van Veldhuizen, J Am Coll Cardiol 2013;61:1498–506, Kang et al, Heart 2015;101:1881–8 en recently also Salah et al, Heart 2019; doi: 10.1136/heartjnl-2018-314173 Prognosis and NT-proBNP in heart failure patients with preserved versus reduced ejection fraction.)
How can this discrepancy be explained ? Mortality is predicted by natriuretic peptides, similarly for HFpEF as for HFrEF, but not only by natriuretic peptides, and the risk in HFrEF of dying may therefore be somewhat higher than in HFpEF despite similar natriuretic peptides, but it also may be lower in HFrEF, than in HFpEF, depending on mortality risks from other diseases.
2. Diagnosis of HFpEF does not fully depend on natriuretic peptides in the ESC guideline (subheading 4, paragraphs on Implications of natriuretic peptides inn heart failure with preserved ejection fractin diagnosis, line 187); when the levels are too low, the diagnosis is not recommended by the ESC, but cannot be excluded yet. As an excellent review on this topic already mentioned (Zakeri R, Cowie MR. Heart 2018;104:377–384): "The gold standard to confirm (or refute) a diagnosis of HFpEF is based on demonstration of elevated LV filling pressures during cardiac catheterisation".
In this respect, also the comments on clinical trials (line 222) using different cut-off criteria is not a proper comment: these cut-off criteria for the trials are not the same as using diagnostic criteria for HFpEF. These cut-off criteria were used to select a group of HFpEF patients that were deemed suitable for these studies (with a high enough risk). So lines 227-230 are not correct.
There is also an incomplete understanding what natriuretic peptides may do in HFpEF or what relationship there is between the severity of HFpEF and the benefits from therapies (MRA's):
1. The paragraph 5, "Implications of natriuretic peptides in .. Therapy": mentions RAAS inhibition, but does not mention cGMP increases, which may be more important for HFpEF patients, since they have reduced cGMP formation which can not be improved by PDE5 inhibitors. This shortage in cGMP may however be improved by neprilysin, by increasing the NP levels. The authors did however mention in Line 289 that nesiritide increases cGMP, but did not use this information earlier, in more general way at the beginning of the paragraph or in paragraph 3, where other molecular effects are described.
See for an excellent review on this topic, Zakeri R, Cowie MR. Heart 2018;104:377–384, see pages 380 + 382 figure 3). For basic understanding: Paulus WJ, Tschöpe C. A novel paradigm for heart failure with preserved ejection fraction: comorbidities drive myocardial dysfunction and remodeling through coronary microvascular endothelial inflammation. J Am Coll Cardiol 2013;62:263–71 and van Heerebeek L, Hamdani N, Falcão-Pires I, et al. Low myocardial protein kinase G activity in heart failure with preserved ejection fraction. Circulation 2012;126:830–9.
2. The benefit of MRA's on HFpEF has been tested in TOPCAT and Aldo_DHF and have been tested against the baseline natriuretic peptide levels. For TOPCAT these interactions have been described in your reference 84 (which you did not use): Interaction Between Spironolactone and Natriuretic Peptides in Patients With Heart Failure and Preserved Ejection Fraction: From the TOPCAT Trial. Anand IS, Claggett B, Liu J, Shah AM, Rector TS, Shah SJ, Desai AS, O'Meara E, Fleg JL, Pfeffer MA, et al. JACC Heart Fail. 2017 Apr; 5(4):241-252. Their conclusion: " Similar to the effects of irbesartan in the I-PRESERVE (Irbesartan in Heart Failure With Preserved Ejection Fraction) trial, a greater benefit of spironolactone was observed in the group with lower levels of NPs and overall risk in TOPCAT. Elevated NPs in HFpEF identify patients at higher risk for events but who may be less responsive to treatment".
So here is something that is worth thinking about, is it increased fibrosis that higher natriuretic peptide levels disclose? And does a MRA work when it can prevent such fibrosis in patients with still low levels of natriuretic peptides ?
Author Response
Please find enclosed the attached word document with the point-by-point responses.

Reviewer 2 Report
This review article is well-written, and the diagnostic, prognostic and therapeutic roles of NPs in HFpEF patients as well as HFrEF patients are systematically described. It may be more helpful if the authors add the information of various ANP and BNP molecular forms including glycosylated proBNP (Matsuo et al. Peptides. 2019;111:3-17). Throughout the paper “plasma” and “serum” are not clearly separated. ANP and BNP cannot maintain stability in serum mainly due to degradation by neural endopeptidase, thus plasma ANP and BNP levels are usually used in the clinical setting. Conversely, the high stability of NT-proBNP makes it possible to measure with serum.
Author Response
Please find enclosed the word document containing the point-by-point responses.

Round 2
Reviewer 1 Report
manuscript (ID: ijms-490663) "Implications of Natriuretic Peptides in Heart Failure with Preserved Left Ventricular Ejection Fraction: from Molecular Evidences to Clinical Implications".
Reviewer 1
Title: is improved.
to "Natriuretic peptides in heart failure with preserved left ventricular ejection fraction: from molecular evidences to clinical implications".
Paragraph 3 to make a figure of the NP effects on downstream molecular mechanisms (RAAS, cGMP) via the receptors. Is a nice figure.
I think that nitrite is also a NO donor and therapies have started: see Chirinos et al, The Nitrate-nitrite-NO pathway and its implications for Heart Failure and Preserved Ejection Fraction. Curr Heart Fail Rep. 2016 Feb; 13(1): 47–59. See also Zamani P, Rawat D, Shiva-Kumar P, Geraci S, Bhuva R, Konda P, Doulias PT, Ischiropoulos H, Townsend RR, Margulies KB, et al. Circulation. 2015 Jan 27; 131(4):371-80. You do not have to add the references, you can add the term ‘nitrite’ below nitrates.
Paragraph 5 to incorporate the BNP analogs nesiritide and carperitide in Table 2. Please make a proper distinction in this table in the effects that are observed specifically in HFpEF patients; They were added, and new reference 61 illustrates HFpEF specific effects of nesiritide. Fine.
And we have introduced the following lines: "The use of recombinant NPs in acute HF is recommended by ESC [3], but without any distinction regarding EF.". Fine
At this time, I can only report the larger comments, and in general, the text is too extensive; it may be shortened with ease, if you can focus on the main messages. The text may be interesting for the reader after major revision. We agree that the text is extensive; we found it difficult to discuss about HFpEF without taking into consideration HFrEF, especially in what concerns therapy. Even in the ESC guidelines, data on HFpEF patients are frequently related and compared to their reduced EF counterparts and current research focusing on this topic is vast and rapidly growing and we desired to not neglect it. Therefore, we kindly ask you to allow us to maintain the actual length of the information included in the manuscript, taking into consideration your valuable suggestions, remarks and overall contribution to improvement the quality of this review. We thank you very much.
Reviewer 1:
I can see the difficulty of shortening the text, however be considerate for the readers with too much information that does not help to understand the main ideas.
I would ask the authors to adapt the heading text of section 5 “Implications of natriuretic peptides.. Therapy” to: 5. “Therapeutic implications of natriuretic peptides .. “, and in section 6 “Implications of natriuretic peptides … Prognosis” to 6. Prognostic implications of natriuretic peptides …”. This gives a better direct reading.
In addition, reconsider placing subheadings in these sections, as in section 5: 5.1 Natriuretic peptide analogs, 5.2 Natriuretic peptide breakdown inhibitors”.
You can shorten the text in section 5
By removing text lines 358-61, “In regards with chronic 359 treatment, a study randomized HF patients to 8 weeks of subcutaneous nesiritide, with a reduction 360 in LAVI, E/e’ and end-systolic and diastolic LV volumes [505046]. However, this study included only 361 patients with an LVEF< 35%.”
by reducing text between lines 408 – 474, in which you mention the results of PARADIGM (Line 408), and discuss whether BNP > 150 etc should be indications for treatment in PARADIGM, which has no bearing on the later HFpEF discussion. So this information can be left out. The appropriate discussion would starting at line 447 in which you discuss why these ARNI’s would be beneficial for cGMP and then with the actual PARAMOUNT and PARAGON studies. The point that you want to make, is whether you may use ARNI for treatment of HFpEF, and the side-point is that maybe some criticism is appropriate as to the inclusion criteria. I advise to start at line 447, skip PARADIGM, and insert the discussion with PARAMOUNT and PARAGON, and add some necessary information (side effects) from the PARADIGM later in this chapter, if necessary.
You can shorten the text in section 6
Line 504-516 start a discussion on levels of NP’s and causes of these levels, but confirms that lower levels are prognostically good and higher levels not. This whole paragraph is summarized by the previous sentence in line 502-504, “It is agreed upon that although NPs serum values tend to be lower in HFpEF patients with no 503 consensus regarding diagnostic thresholds, they retain their prognostic utility irrespective of LVEF. So, you may remove 504-16, and add reference of TOPCAT to the line 504 with reference 3.
It seems that you do not yet understand that NP’s give a relative risk of prognosis, and not an absolute risk. So, the statement that mortality is lower in HFPEF patients than in HFREF patients in the study by Salah, while NP’s are similarly predictive, can still be true. It has to be stated, that in Salah’s paper the mortality was not significantly lower in HFpEF than in HFrEF. But in fact, it does not really matter, HFpEF may even have increased mortality compared to HFrEF in some studies, but the NP’s still will show the same relative risks for HFpEF and HFrEF. The mortality differences are then explained by other causes, also giving relative risks.
You discuss the ‘tresholds’ of NP’s for prognosis (line 539-58), but you already have the information from the paper of Levy and Anand (line 519-21) that using prognostic tresholds is not necessary, or helpful because you can assess prognosis continuously. You can try to find a threshold, of course, like the threshold of 1000 ng/L NTproBNP that was presented as a risk treshold for HFrEF, but if you do not succeed, please keep this paragraph limited by simply stating that different possible NP risk tresholds have been presented with some references.
Line 559-569 about the risk of Afib predicted by NP’s. You may shorten these sentence to one sentence, as Afib does not impact on prognosis, but a remark about it is interesting of course.
Abstract line 34 and introduction line 52: HFpEF "patients share the same prognosis as those with reduced ejection fraction" . We agree that the message was lacking clarity regarding HFpEF patients’ prognosis. We have changed lines 34 (abstract) and 52 (introduction) according to your suggestions.
Abstract: "It seems that patients have a better prognosis when compared to those with reduced ejection fraction" and Line 52: here the sentence was removed.
Line 52: "This increasing incidence justifies the need for proper diagnostic, therapeutic and prognostic tools." Is OK.
The evidence for a reduced or similar prognosis in HFpEF versus HFrEF is both available; in most acute HF studies, the prognosis is similar, while in recent chronic HF studies the prognosis is better for HFpEF than for HFrEF. So I would change this sentence in Abstract again to : “HFpEF patients either share the same prognosis or have a better prognosis compared to those with reduced ejection fraction”.
We have added references 95-100:
95.Berry, C.; Doughty, R.N.; Granger, C.; et al. The survival of patients with heart failure with preserved or reduced left ventricular ejection fraction: an individual patient data meta-analysis. Eur Heart J. 2012, 33, 1750-7. doi: 10.1093/eurheartj/ehr254
96. Lam, C.S.P.; Gamble, G.D.; Ling, L.H.; Sim, D.; Leong, K.T.G.; Yeo, P.S.D.; Ong, H.Y.; Jaufeerally, F.; Ng, T.P.; Cameron, V.A.; Poppe, K.; Lund, M.; Devlin, G.; Troughton, R.; Mark Richards, A.; Doughty, R.N.; Mortality associated with heart failure with preserved vs. reduced ejection fraction in a prospective international multi-ethnic cohort study. Eur Heart J. 2018, 39, 1770-1780. doi: 10.1093/eurheartj/ehy005
97. Levy, W.C.; Anand, I.S.; Heart failure risk prediction models: what have we learned? JACC Heart Fail. 2014, 2, 437-9. doi: 10.1016/j.jchf.2014.05.006
98. van Veldhuisen, D.J.; Linssen, G.C.; Jaarsma, T.; van Gilst, W.H.; Hoes, A.W.; Tijssen, J.G.; Paulus, W.J.; Voors, A.A.; Hillege, H.L. B-type natriuretic peptide and prognosis in heart failure patients with preserved and reduced ejection fraction. J Am Coll Cardiol. 2013, 61, 1498-506. doi: 10.1016/j.jacc.2012
99. Kang, S.H.; Park, J.J.; Choi, D.J.; Yoon, C.H.; Oh, I.Y.; Kang, S.M.; Yoo, B.S.; Jeon, E.S.; Kim, J.J.; Cho, M.C.; Chae, S.C.; Ryu, K.H.; Oh, B.H. Prognostic value of NT-proBNP in heart failure with preserved versus reduced EF. Heart. 2015, 101, 1881-8. doi: 10.1136/heartjnl-2015-307782
100. Salah, K.; Stienen, S.; Pinto, Y.M.; Eurlings, L.W.; Metra, M.; Bayes-Genis, A.; Verdiani, V.; Tijssen, J.G.P.; Kok, W.E. Prognosis and NT-proBNP in heart failure patients with preserved versus reduced ejection fraction. Heart. 2019. pii: heartjnl-2018-314173. doi: 10.1136/heartjnl-2018-314173. [Epub ahead of print]
We are very sorry for this misleading piece of information related to HFpEF patients’ prognosis. We changed our statement on this issue. We have added the following paragraphs regarding the prognostic role of NPs in HFpEF, emphasizing that these patients have lower mortality rates as compared to HFrEF, but at a given NT-proBNP levels they share the same mortality HR, in part due to increased comorbidities and different mechanisms leading to NP elevation between HF phenotypes.
No, the NP levels share the same HR for mortality in HFpEF and HFrEF, while the mortality that is related to other causes is simply not seen in these NP levels, and thus comorbidities are also related to mortality, but are distributed differently among HFrEF and HFpEF patients. The main issue here is that NP’s should not be seen as (all explaining) surrogate markers of mortality.
“The NPs concentrations retain their prognostic utility albeit their lower baseline levels in HFpEF patients [95]. Several studies reported similar death risks for a given NPs concentration irrespective of HF phenotype [95-100]. Levy and Anand compared patients from I-PRESERVE and VALHEFT trials, showing that a 1-log increase in NT-proBNP levels carries a mortality HR of 1.7 regardless of LVEF [97]. However, the same authors demonstrated that the mortality of HFrEF patients was two thirds higher than of those with HFpEF. Similarly, Salah et al. reported that hospitalized HFpEF patients have lower mortality rates when compared to HFrEF [100]; however the risk of death tends to equalize after discharge. More importantly, although patients with HFpEF display lower baseline NPs concentrations, for the same NT-proBNP level, prognosis is similar regardless of LVEF. This paradox can be firstly be explained by the different mechanisms involved in NPs secretion between the two phenotypes. It is known that at the same end-diastolic LV pressure, HFpEF patients display lower NT-proBNP levels. This may be due to NPs correlation with diastolic wall stress which, according to the law of La Place, correlates with wall pressure and cavity diameter and is inversely related to wall thickness. As patients with HFpEF typically exhibit a concentric LV remodelling with increased wall thickness, this may explain in part their lower NPs levels for a given increased wedge pressure as compared to the HFrEF patients (who typically have an eccentrical LV remodelling). Secondly, this paradox can be explained by the distribution of comorbidities in regard with HF phenotype. As such, HFpEF patients tend to be older, with increased incidence of arterial hypertension, chronic kidney disease, AF and anemia, comorbidities that may account for similar prognosis between HF phenotypes despite lower NTpro-BNP of HFpEF [96,100]. This drives attention to the need of also addressing non-cardiovascular diseases in order to improve outcomes in HFpEF patients.”
Diagnosis of HFpEF does not fully depend on natriuretic peptides in the ESC guideline (subheading 4, paragraphs on Implications of natriuretic peptides inn heart failure with preserved ejection fractin diagnosis, line 187); when the levels are too low, the diagnosis is not recommended by the ESC, but cannot be excluded yet. As an excellent review on this topic already mentioned (Zakeri R, Cowie MR. Heart 2018;104:377–384): "The gold standard to confirm (or refute) a diagnosis of HFpEF is based on demonstration of elevated LV filling pressures during cardiac catheterisation".
We appreciate your remarks on the integrative modality of HFpEF diagnosis. Therefore, we have introduced reference 26:
Zakeri, R.; Cowie, M.R. Heart failure with preserved ejection fraction: controversies, challenges and future directions. Heart. 2018, 104, 377-384. doi: 10.1136/heartjnl-2016-310790
We have also added the following regarding HFpEF diagnosis:
“Diagnosing HFpEF remains difficult due to a lack of consensus, patients’ heterogeneity and multiple concurrent pathologies that may mimic not only HF symptoms, but also lead to either increased (AF) or decreased (obesity) NP levels [26]. HFpEF diagnosis requires clinical and imagistic criteria, as well as elevated NPs levels. Given that one third of HFpEF have normal NPs levels [26], relying solely on their values for diagnosis is not recommended and their levels must be interpreted in the clinical context. As such, the diagnostic gold standard is cardiac catheterization showing increased LV filling pressures [26].”
Reviewer 1: Fine
In this respect, also the comments on clinical trials (line 222) using different cut-off criteria is not a proper comment: these cut-off criteria for the trials are not the same as using diagnostic criteria for HFpEF. These cut-off criteria were used to select a group of HFpEF patients that were deemed suitable for these studies (with a high enough risk). So lines 227-230 are not correct.
Thank you very much for your suggestions. We have deleted lines 222 and 227-230.
We have reformulated the paragraph on the use of different NPs thresholds for sinus rhythm and AF patients in clinical trials: "In the literature, several studies report the use of different NPs thresholds with respect to underlying cardiac rhythm (sinus versus atrial fibrillation): 600 pg/mL in the SOCRATES trial [39] and >900 pg/mL in PARAGON trial [40]."
Reviewer 1: Fine
The paragraph 5, "Implications of natriuretic peptides in .. Therapy": mentions RAAS inhibition, but does not mention cGMP increases, which may be more important for HFpEF patients, since they have reduced cGMP formation which can not be improved by PDE5 inhibitors. This shortage in cGMP may however be improved by neprilysin, by increasing the NP levels. The authors did however mention in Line 289 that nesiritide increases cGMP, but did not use this information earlier, in more general way at the beginning of the paragraph or in paragraph 3, where other molecular effects are described.
See for an excellent review on this topic, Zakeri R, Cowie MR. Heart 2018;104:377–384, see pages 380 + 382 figure 3). For basic understanding: Paulus WJ, Tschöpe C. A novel paradigm for heart failure with preserved ejection fraction: comorbidities drive myocardial dysfunction and remodeling through coronary microvascular endothelial inflammation. J Am Coll Cardiol 2013;62:263–71 and van Heerebeek L, Hamdani N, Falcão-Pires I, et al. Low myocardial protein kinase G activity in heart failure with preserved ejection fraction. Circulation 2012;126:830–9.
We thank you very much for your recommendations. As such, we have introduced references 24-25:
We have introduced the following paragraphs regarding cGMP/PKG/NPR pathway (paragraph 3):
“As a second messenger, cGMP is formed from 2 possible precursors, either soluble guanylyl cyclase- sGC (found in cytosol; it requires nitric oxide binding) and particulate guanylyl cyclase (pGC), found in the cellular membrane and activated via NPR. As such, this activation leads to an increase in cGMP, which in turn increases protein-kinase G (PKG) levels [24-26]. The latter phosphorylates several proteins, including myocardial cytoskeletal titin [25]. Moreover, decreased levels of cGMP and subsequently of PKG have been associated with myocardial remodeling through increased cardiomyocyte hypertrophy and resting tension [25] (figure 1). HTN, AF, chronic kidney disease- frequently found comorbidities in HFPEF patients, determine a decrease in cGMP through a pro-inflammatory state and subsequent decrease of nitric oxide. Importantly, augmenting cGMP concentrations may constitute therapeutic targets in HFpEF.”
Reviewer 1: Fine
For the implication of cGMP in HFpEF therapy we have introduced in paragraph 5 (Therapy):
“Regarding cGMP and sacubitril/valsartan therapy, there is extensive focus on its beneficial effects on cGMP levels and, subsequently, PKG. cGMP deficiency is of paramount importance in HFpEF, as it lowers PKG levels and thus promotes myocardial remodeling (both hypertrophy and impaired relaxation through increased cardiomyocyte resting tension) [24]. Through oxidative stress [25] and subsequent decrease in NO availability [24], numerous comorbidities contribute to the lower levels of cGMP found in HFpEF. It seems that the pro-inflammatory state induced by these comorbidities triggers coronary microvascular inflammation, which further decreases NO availability [24] and inhibits cGMP formation. Moreover, oxidative stress affects titin, a protein found in cardiomyocytes cytoskeleton in two ways: firstly by inhibiting PKG- dependent phosphorylation and secondly by determining the formation of disulfide bridges, which leads to a more stiffer titin molecule [24]. Authors agree that the cGMP deficiency in HFpEF is decreased production and not increased breakdown [23], which explains why in HFpEF patients, cGMP levels increase in response to neprylisin inhibitors and not phosphodiesterase inhibitors, which inhibit cGMP breakdown [25].”
Reviewer 1: Fine
The benefit of MRA's on HFpEF has been tested in TOPCAT and Aldo_DHF and have been tested against the baseline natriuretic peptide levels. For TOPCAT these interactions have been described in your reference 84 (which you did not use): Interaction between Spironolactone and Natriuretic Peptides in Patients With Heart Failure and Preserved Ejection Fraction: From the TOPCAT Trial. Anand IS, Claggett B, Liu J, Shah AM, Rector TS, Shah SJ, Desai AS, O'Meara E, Fleg JL, Pfeffer MA, et al. JACC Heart Fail. 2017 Apr; 5(4):241-252. Their conclusion: "Similar to the effects of irbesartan in the I-PRESERVE (Irbesartan in Heart Failure With Preserved Ejection Fraction) trial, a greater benefit of spironolactone was observed in the group with lower levels of NPs and overall risk in TOPCAT. Elevated NPs in HFpEF identify patients at higher risk for events but who may be less responsive to treatment".
So here is something that is worth thinking about, is it increased fibrosis that higher natriuretic peptide levels disclose? And does a MRA work when it can prevent such fibrosis in patients with still low levels of natriuretic peptides?
We thank you very much for your suggestion.
We are very sorry for this error. We have now cited reference 84 (currently 89, after the addition of other references) and added references 90-92:
Reviewr 1: Fine
Indeed, the fact that high-risk HFpEF patients with higher NPs levels are the ones who would most benefit from specific therapeutic interventions is uncertain. We have added the following paragraphs regarding emphasizing the relation between NPs and myocardial fibrosis and the benefit of MRAs therapy in HFpEF patients with lower NPs levels.
“Apart from novel therapies, several studies investigated the benefits of mineralocorticoid receptors antagonists (MRAs) therapy in HFpEF [6,89-92].The sub analysis of TOPCAT trial conducted by Anand et al. reveals that while spironolactone does not impact overall mortality and hospitalization rates, it proved beneficial in HFpEF patients with lower NPs levels and not in those with increased concentrations [89]. These results were similar to those of the I-PRESERVE trial [1], in which Irbesartan treatment was beneficial in HFpEF patients with lower NPs levels. The results of the two trials point out the fact that higher NPs levels in HFpEF translate an increased cardiovascular event rate, not necessarily increased treatment responsiveness. Moreover, it seems that higher NPs levels in these patients relate to a more advanced degree of structural heart disease, including higher degree of myocardial fibrosis, less responsive to therapeutic intervention [89].
Patients with HFpEF present increased fibrosis biomarkers, as shown by Zile et al. [90] and Oikonomou et al. [6] In a study conducted by Cho-Kai Wu et al. [91], HFpEF patients with increased myocardial fibrosis degree as assessed by late-gadolinium enhancement magnetic resonance imaging scans had higher levels of NT-proBNP. Fibrosis biomarkers concentrations (growth differentiation factor, galectin-3, tissue inhibitor of metalloproteinase, matrix metalloproteinase 2) correlated with the severity of fibrosis. However, the authors concluded that NPs rather reflect wall tension and volume overload with lower discriminative power in regard to the presence and/or degree of myocardial fibrosis.
It appears that HFpEF with lower matrix metalloproteinase 9 levels benefit most from eplerenone treatment [6]. Chen et al [92] show that MRA therapy in HFpEF reduce both myocardial remodeling and amino terminal peptide of procollagen III levels, without however affecting NP levels. It may be that in early stages of HFpEF, the progression of myocardial remodeling and subsequent fibrosis is preventable by therapeutic interventions, such as MRAs therapy, thus explaining the benefit that patients with lower fibrosis biomarkers and NPs levels have from this therapy.”
Reviewer 1: agreed
Comment on Abstract
The abstract does not reflect the contents of the paper well. It takes on a discussion on diagnostic tresholds and prognostic tresholds, but this is not the real message of the paper, is it ? This can be improved, in which simply mentioning the highlights from the sections is asked for.
Line 239: “The longer plasma half-life of BNP and NT-proBNP (22 minutes and 70 minutes, respectively) as compared to ANP (2 minutes), makes the two the gold standard”: this is not really the case. Gold standard in diagnosis is still the pressure measurement, and in reality, the use of NP’s is more effective in EXCLUDING heart failure. This has been challenged for HFpEF, and therefore you will use the golden standard left ventricular pressure in case of doubt. You already adopted this approach in lines 236-37 with reference 26. In case of elevated levels, you have diagnostic uncertainty (no large PPV) and you will need additional imaging to confirm the diagnosis.
In addition, limited evidence is there for ‘guiding’ therapy in HFpEF with NP’s, so also here no golden standard. What you mean is probably: NP’s are very useful for diagnosis and possibly also for guiding of therapies.
Line 248-57: “Given that NPs serum levels are affected by several factors, including the presence of AF and body mass index, their use rather resides in excluding the diagnosis of HF with a subsequent high-negative predictive value (0.94-0.98) [3]. As such, especially in what concerns the diagnosis of HFpEF, NPs serum levels must be corroborated with both clinical context and other imagistic parameters”. I would agree, on both ends of the spectrum: in those with too low NP levels and still signs of HFpEF, and in those with elevated levels. The ESC guideline has given these NP levels as diagnostic tresholds, as not to INCLUDE everyone with signs of heart failure as HFpEF but have at least the criterium of elevated NP. Still, in its text, it is possible to diagnose HFpEF without NP’s.
Author Response
Please find enclosed the response to the reviewer's suggestions.
